# The LNG Flow Simulation in Stationary Conditions through a Pipeline with Various Types of Insulating Coating

Ildar Shammazov and Ekaterina Karyakina *

The Department of Transportation and Storage of Oil and Gas, Saint Petersburg Mining University, Saint Petersburg 199106, Russia
* Correspondence: s195079@stud.spmi.ru

**Abstract:** Liquefied natural gas (LNG) is one of the most promising fuels for energy supply because it has a favorable combination of environmental and economic properties in connection with new trends aimed at the development of ecological and sustainable consumption of natural resources, which ensure a constant growth in LNG consumption. The article presents an analytical review of the main technical solutions for the construction of cryogenic pipelines and insulating coating structures. The ANSYS Fluent software was used for simulation of the LNG flow in a pipeline section 10 m long with an outer diameter of 108 mm for three types of insulating coating (polyurethane (PU) foam, aerogel, and vacuum-insulated pipe (VIP)). In addition, an assessment was made of the insulating effect on the LNG temperature distribution along the length of the pipeline. The largest increase in temperature from 113 K to 113.61 K occurs in PU foam-insulated pipes; the smallest was observed in VIP. Further, as an alternative to steel, the use of ultra-high molecular weight polyethylene (UHMWPE) for pipeline material was considered. The optimal result in terms of temperature distributions was obtained while simulating the flow of an LNG pipeline with PU foam by increasing the thickness of the insulating coating to 0.05 m.

**Keywords:** thermal insulation system; vacuum-insulated pipeline; polyurethane foam; aerogel; UHMWPE; CFD simulation





## 1. Introduction

The issue of pipeline transport of liquefied gases has been discussed by scientists since the second half of the 20th century. At that time, several test specimens were built, but most of them were too short, and some of them failed.

The process of LNG transportation is associated with a large number of technical and technological difficulties, the solution to which requires an integrated approach.

The main point that should be taken into account during all stages of the pipeline life cycle, from its design to operation, is the prevention of two-phase flow formation along the entire length of the pipe.

The presence of non-equilibrium states in the phases during cryogenic liquid transportation leads to the occurrence of pressure pulsations and flow destabilization, which eventually can provoke emergency situations.

As an example, LNG overheats and begins to boil due to a high temperature gradient between the medium that enters the filling process and the pipeline itself. The bubbles formed during the boiling process start to grow and subsequently merge into a larger bubble (the term "Taylor bubble" can be found in the literature), which fills the entire section of the pipeline. As it is rising, the "Taylor bubble" pushes the liquid from the filled pipeline back into the storage tank. At this point, the cold liquid at the bottom of the tank rushes into the empty line due to gravity and the reduced pressure created by the condensation of steam in the line. This fluid column impacts a closed valve or any other obstruction at the bottom of the line with a velocity high enough to create a potentially

damaging fluid hammer. The pressure rises sharply, and the liquid is ejected back into the storage. [1–3].

The mathematical description and numerical simulation of phase transitions and two-phase flows, as well as the prediction of the phenomena mentioned above, are very difficult tasks to perform [3–8].

The issues of appropriate material selection for the pipeline and the basic principles of design are very important due to the fact that LNG pipelines operate at cryogenic temperatures.

Additionally, apart from the fact that pipeline material has to have high impact strength and plasticity in combination with high strength [9–11], it is necessary to select the insulating coating of the pipeline correctly, as well as the technology of its application.

The system of LNG production and distribution represents a single technological complex, which also includes LNG transportation and storage systems. At the same time, technologies are being developed that help reduce LNG evaporation during storage [12–14] as well as prediction models for LNG evaporation [15–17]. The latter helps to reduce the risks of flow destabilization on the process lines. In this regard, maintaining the LNG temperature in the production system is achieved both through appropriate material choice and the integration of technological solutions based on the basic principles of fluid dynamics.

Furthermore, much attention is being paid to the development of hydrogen energy and renewable energy sources as an alternative to LNG. However, these sources have a number of unsolved problems that do not allow them to become applicable alternatives to LNG in the foreseeable future.

As a result, the widespread use of hydrogen energy is largely prevented by certain issues with safe storage and transportation, such as fluid evaporation, high corrosivity, low production efficiency, insufficient infrastructure, and a lack of developed mechanisms for market interaction. Furthermore, renewable energy technologies today are unable to ensure the production of energy sufficient to cover the ever-increasing demand for energy resources. [18–20].

The purpose of this article is to analyze the existing designs of cryogenic pipelines used for the transportation of liquefied gases, as well as solutions for the installation of their insulating coating.

## 2. Related Research

### 2.1. World Experience in the Construction of Liquefied Gas Pipelines

There are a large number of design developments for cryogenic pipelines. However, with all their diversity, they can be divided into two broad categories:

- pipe-in-pipe structures;
- flexible cryogenic pipelines.

The following offers an overview of the world's experience in the construction of liquefied gas pipelines.

A large number of cryogenic pipelines [21–24] were developed decades ago. Such pipelines used to consist of a carrier pipe and a casing with layered insulation containing an elastic adsorbent, the space between which was evacuated. Figure 1 shows the general design of such pipelines.

The system developed by FW-FERNWÄRME-TECHNIK GmbH for the transport of liquefied gas on ships, FW LNG-PIPE, also consists of an inner pipe transporting cryogenic liquid, a frost-resistant heat-insulating layer, and an encasing pipe. The inner pipeline is guided by bearings inside the casing, and the annular space is evacuated to 1 mbar. The pipes are thermally insulated with flexible silicate aerogel insulation of the Cryogel type. As part of the development, a 50-m simulation route was built on the territory of the plant. The pipeline executed the transfer of liquid nitrogen at a temperature of −196 °C [25].

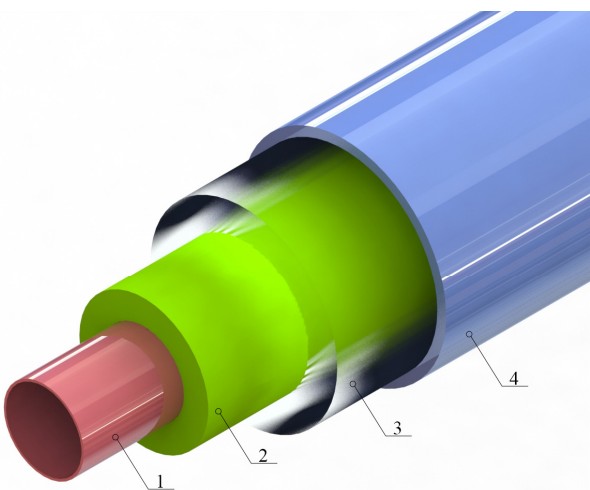

**Figure 1.** Typical design of cryogenic pipeline insulation [21–24]. 1—pipeline; 2—insulation with adsorbent; 3—evacuated space; 4—jacket cover.

ITP Interpipe has developed the world's first fully buried cryogenic pipeline to load ethane and LNG from an existing jetty into a new storage tank (Gujarat Buried Cryopipe) that has been operated since 2017. The pipe-in-pipe system consists of two parallel, fully underground 24-inch (610 mm) pipes made of Invar, an alloy with a 36% nickel content, carrying cryogenic liquid with an estimated transport temperature of −168 °C [26].

The development of pipelines with vacuum insulation by Chart VIP Design Services has found wide application in the LNG industry. Their pipeline consists of the following elements. External stainless-steel pipe provides structural integrity to the annular vacuum. Fiberglass supports keep the inner and outer shapes of the pipeline concentric. The stainless-steel inner line is capable of transporting cryogenic liquids at temperatures down to −268 °C, with operating pressures up to 2.76 MPa. Internal stainless-steel bellows help to absorb thermal expansion, and an insulating vacuum system is applied to minimize convective heat transfer. Additionally, in order to minimize heat losses, layers of cryogenic film and glass paper cover the carrier tube.

The system was first launched for LNG pumping at the Atlantic Train 1 facility in 1998 in Trinidad, West Indies, through a 760 m long LNG cooling line (still in operation).

A similar system has been used since 2006 to pump LNG through 26″ (660 mm) process lines and to loop through a 12″ (305 mm) evaporation line, with a total length of 4.3 km and an operating pressure of 1.54 MPa, at the export terminal Freeport LNG in Texas.

In addition, such a system is also installed at the LNG plant in Darwin, Australia, and has been in continuous operation since 2005 with a total length of 6.4 km and carrier pipe diameters ranging from 4 inches (100 mm) to 30 inches (762 mm) [27].

The design of a flexible cryogenic pipeline, patented in [20,28], contains an inner pipe (located in a corrugated external pipe) that transports liquefied natural gas. The inner pipeline consists of cylindrical segments, which are overlapped by successive corrugations of the pipe. In order to ensure fixation of the inner and outer corrugated pipes, the external stop of axial fixation is provided on the inner pipe.

The inner pipe can be made of stainless steel 304L or 316L with a diameter of 300 to 600 mm, while the length of such lines is not more than 300 m.

Such a pipeline is intended for the transport of cryogenic fluids below their normal boiling point. This temperature is a result of the gas liquefaction process on floating installations of the FLNG type.

It is worth noting the manufacturing complexity of this system and the fact that the presence of a large number of elements in the design leads to a decrease in reliability and maintainability. It should also be noted that when creating pipelines for LNG pumping on floating installations, it should be taken into account that the loading/unloading process

is carried out in difficult meteorological conditions and is often associated with large hydraulic losses.

Having analyzed the market for flexible cryogenic pipelines in open sources, the authors found only flexible pipelines of small diameter (not higher than 264 mm), which indicates the current absence of industrial production of flexible cryogenic pipelines. The most likely application of the technology described in [28,29] is within the framework of one particular project.

The technologies for the production, transportation, and storage of LNG are always difficult to implement, as these projects often require large-scale work over many years. One of the challenges is the fact that a number of changes in the current regulatory and legal framework may occur. Another issue is that such projects also require significant sources of funding.

In [25], the authors describe the process of introducing pipeline transport technologies for industrial use. The technologies were developed in the course of a joint industry project. Framo Engineering, together with Aker Pusnes, Kongsberg Oil & Gas Technologies (KOGT), Nexans, and MIB, have developed a tandem LNG loading system based on a technology similar to the tandem loading used to transfer crude oil between two floating vessels. A system of corrugated, flexible pipes made of double-walled stainless steel with vacuum insulation is used to ensure transmission between the two vessels. The pipe is produced in one continuous length up to 150 m without intermediate connections.

The authors of the article [30] mention that such projects are always challenging and require the work of representatives of different companies.

In addition, not having any conflicts of interest is of paramount importance for the companies involved in the project. All participants must be interested in the development of different technologies. However, the final outcomes should be beneficial for every company taking part in the project.

It should also be noted that LNG projects are almost always unique developments of certain companies that prefer not to disclose them, which, in a certain sense, hinders the development of the industry as a whole. In this regard, there is a growing interest in small-scale LNG projects to provide flexible and decentralized energy supply systems [31–33].

In addition, the data on the long-term operation of cryogenic LNG pipelines is scarce, especially on the changes in the strength characteristics of the pipe. In [34], the authors present an experimental study of changes in the strength characteristics of a pipeline that had been in operation for 40 years (since 1964). The pipeline was made of 304 stainless steel for connecting LNG tanks at the Barcelona plant. Ultrasonic testing, chemical analysis, metallography, and tensile strength tests were carried out in order to determine the characteristics of the pipes. The authors did not observe embrittlement or damage to the pipe material at the visual or microstructural levels during the operation.

Apparently, such high rates of strength characteristics during long-term operation are associated with the high capacity of the above steel for structural yielding deformation, as well as the technological process of performing welding work carried out at a high level, which did not lead to the emergence of additional stress concentration zones.

Thus, the issues of material performance for the construction of cryogenic pipelines and the selection of an insulating coating are fundamental to the construction of process lines.

### 2.2. Insulation of Cryogenic Pipelines

In order to maintain LNG in a liquid state and prevent the occurrence of a two-phase flow, as well as to minimize the heat exchange with the environment and heat gains, it is necessary to use complex thermal insulation [35,36].

There are several key points to consider when designing an insulating coating for LNG pipelines.

Due to the fact that any material deforms when exposed to temperature, working with cryogenic temperatures is not an exception and can provoke a change in the dimensions of

the carrier pipe, which can lead to deformation of the insulating coating and the appearance of gaps. In addition, in combination with high humidity, there is a possibility of ice formation, so ensuring dimensional stability is an important parameter of the insulating coating that must be taken into account.

The chances of physical damage to the insulation coating and the product pipe can be significantly reduced by using high compressive-strength insulation materials to absorb point loads and movement. In addition, pipe supports (in the case of above-ground installations) require high compressive strength, since weight loads can damage the insulation in these areas.

Furthermore, it is necessary to pay close attention to the compliance of the produced insulating coating with the parameters of the project, such as dimensions, coating technology with minimization joints perpendicular to the pipe that pass through the entire thickness of the insulating pipe segment and their sealing [36], and the current regulatory standards of various countries.

According to [37], the design of thermal insulation for surfaces with below-ambient temperatures consists of several layers (heat-insulating, vapor-insulating, covering) and fixing elements that do not degrade the insulating properties of the material.

For the heat-insulating layer of equipment and pipelines, it is necessary to use heat-insulating materials with a density of not more than 200 kg/m$^3$ and a design thermal conductivity of not more than 0.04 W/m·K when working with environments below −40 °C. Figure 2 shows a typical insulating coating scheme for surfaces, as mentioned above.

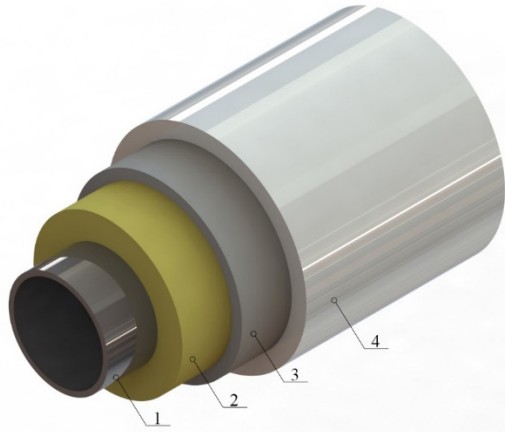

**Figure 2.** Typical scheme of "cold" insulation according to [37]. 1—cryogenic pipeline; 2—heat-insulating layer; 3—vapor barrier layer; 4—cover layer.

The main types of insulation material that are widely used can be divided into several groups: rigid materials (polyurethane foam), vacuum insulation (MLI, superinsulation), and powder (perlite, aerogel).

Polyurethane foam (PUR) is one of the most popular insulating materials used at cryogenic temperatures. Its main advantages include low thermal conductivity, structural strength, light weight, stability (measurement stability is ensured by a low thermal expansion coefficient), low absorption, and high chemical resistance.

Rigid polyurethane foam is compatible with a wide range of support materials, including paper, foil, fiberglass, aluminum, and bitumen. In addition, proper foil selection improves the insulating properties of PU foam by providing protective moisture barriers useful in high humidity environments. This type of insulation is used in a wide range of operating temperatures, from −200 °C to +130 °C.

Figure 3a shows a diagram of the insulating coating of a cryogenic pipeline, which is a combination of rigid materials with cryogel applied on top. A similar solution has been applied at the Canaport LNG terminal in Canada.

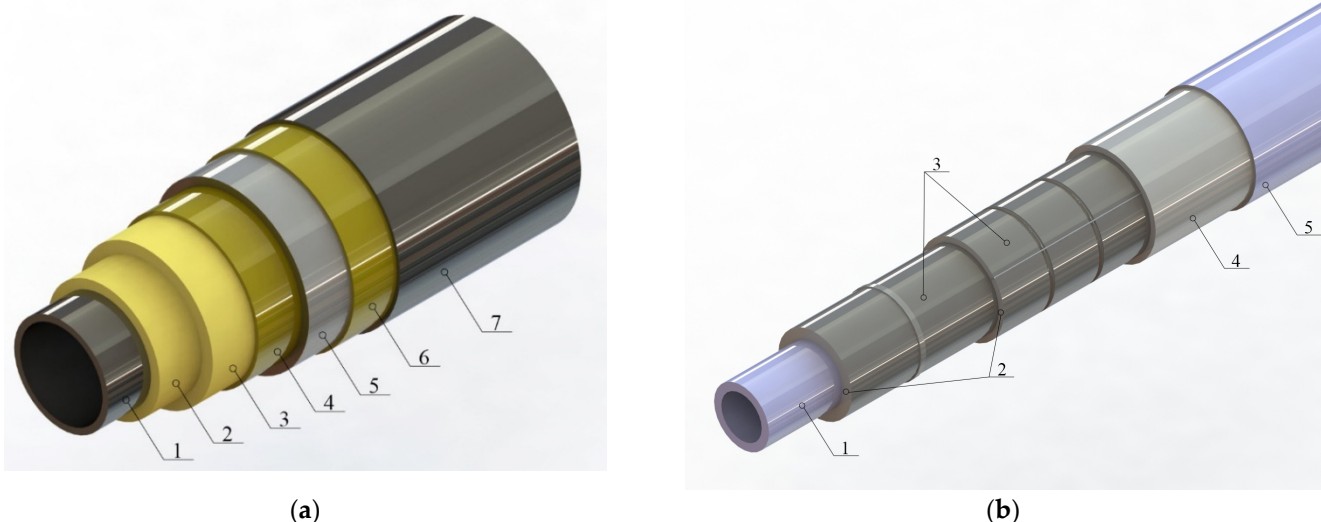

(**a**)　　　　　　　　　　　　　　　　　　　　　　　　(**b**)

**Figure 3.** Examples of cryogenic pipeline insulation systems [38]. (**a**) PIR-Cryogel® Z insulation system. 1—Carrier pipe; 2—PIR; 3—vapor barrier layer of polyester film; 4—layer of Cryogel® Z; 5—butyl-rubber sealing tape; 6—the main vapor barrier layer; 7—protective cover made of carbon steel; (**b**) Typical scheme of insulation of cryogenic LNG pipelines with solid materials—foamed glass. 1—cryogenic pipeline; 2—heat-insulating layers of foamed glass; 3—layers of rolled fiberglass; 4—vapor barrier layer; 5—metal shell.

Figure 3b shows a typical foam glass insulation for LNG pipelines [38].

Among the disadvantages of such insulating materials is the presence of foaming agents that diffuse over time, leading to a change in their thermal characteristics.

Aerogels are composed of light, solid silica particles derived from a gel in which the liquid component has been replaced by a gas. Silica solid particles are poor conductors, consisting of very small, three-dimensional, intertwined clusters that make up only 3% of the volume. Therefore, the conductivity through a solid is very low. The remaining 97% of the volume is filled with air trapped in nanopores. Lack of sufficient space for air circulation hinders convection and gas-phase conduction.

Aerogels are flexible materials that deform when compressed. They can withstand high-impact loads without sacrificing performance. Aerogel applications include cryogenic storage, LNG import/export pipelines, and process areas. Operating temperatures range from −270 °C to 90 °C.

Aerogels have the lowest thermal conductivity of all materials used for cryogenic work. Therefore, they are much thinner in comparison with other "cold" insulation materials. The minimal thickness of aerogels results in a smaller surface area and reduced heat gain compared to other insulating materials. An example of an insulating coating pipe based on aerogel is depicted in Figure 4 [36,39].

Vacuum insulation (superinsulation, multi-layer insulation (MLI), and VIP) is based on the principle of exclusion of convective heat transfer between warm and cold walls, which means that only heat transfer by radiation and the thermal conductivity of residual gases remains [36,40–43]. Such insulation consists of multiple protective screens arranged in parallel as close to each other as possible. Superinsulation usually contains about 25 layers per cm. Each layer is insulated from the other with a spacer material such as polyester, nylon, or mylar. The aluminum foil is carefully wrapped around the pipe so that it covers the entire surface. A separating material is placed between the layers to prevent individual foil coatings from touching each other. If they come into direct contact, a thermal short circuit will occur, and heat transfer will increase. The larger number of layers leads to an increased insulation ability of the system. Typically, layers having an overall thickness of about one inch (25.4 mm) are applied within the liquid nitrogen temperature range described.

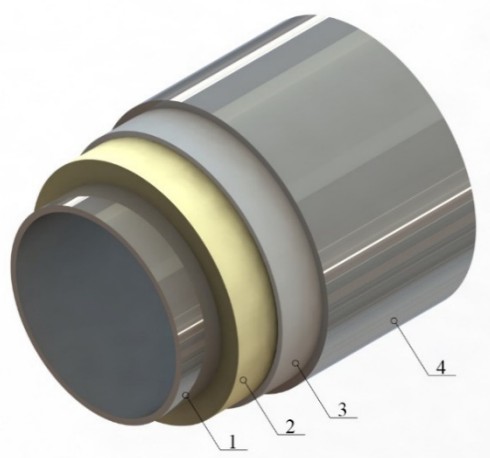

**Figure 4.** Insulating coating based on aerogel [36,38]. 1—Carrier pipe (steel ASTM 316L SS, ASTM 333 Grade 8, AISI 304); 2—A layer of insulating coating of aerogel; 3—Protective jacket pipe made of carbon steel; 4—Concrete coating.

The multilayer insulation is designed to operate in a vacuum at pressures approximately below 0.013 MPa. Production of such a vacuum typically requires long evacuations along with heat and purge cycles. Chemical materials are needed to absorb the released gases in order to maintain vacuum for long periods of time [36,42].

Thus, despite the obvious effectiveness of VIP, it has a number of disadvantages. First of all, the main problem is that the material of such insulation is anisotropic in nature, which makes it difficult to use it for complex geometries. Additionally, this material is very sensitive to mechanical compression and edge effects, thus requiring great care and attention to detail at all stages of its installation. Consequently, its real-life performance is usually worse than that predicted. Moreover, the complexity of creating and maintaining a vacuum along the entire length of the pipeline leads to the limited length and diameter possible for such pipelines.

## 3. Materials and Methods

As part of the analysis of the influence of various types of insulating coating on the temperature distribution along the length of the pipeline, the ANSYS Fluent software package was used to simulate the flow of liquefied natural gas through the pipeline with various types of insulating coating.

The material of the pipe is stainless steel AISI 321, external diameter is $d_{ext}$ = 108 mm; $\delta$ = 5 mm. The initial data for modeling are presented in Tables 1 and 2. The LNG composition was chosen based on the composition ($CH_4$—99.8%; $C_2H_6$—0.07%; $N_2$—0.13%) presented in [44]. The REFPROP 9.0 software was used to calculate the main properties of LNG at the transportation temperature. It can also be found in the reference data in [45] or calculated based on widely-used equations of state [8,46].

**Table 1.** Properties of LNG [44].

| Parameter | Value |
|---|---|
| LNG inlet temperature (T, K) | 113 |
| Density ($\rho$, kg/m$^3$) | 421.35 |
| Heat capacity (Cp, J/kg·K) | 3480.6 |
| Thermal conductivity coefficient ($\lambda$, W/m·K) | 0.18238 |
| Viscosity ($\nu$, Pa·s) | 0.00011451 |
| Mass flow rate (G, t/day) | 500 |
| Gauge pressure (inlet) ($p_1$, MPa) | 0.6 |
| Saturation pressure ($p_s$, MPa) | 0.12 |
| Phase transition temperature ($T_p$, K) | 138.55 |

To determine the parameters of LNG transportation (pressure and temperature), it is generally accepted to use a liquid phase diagram in order to prevent the two-phase flow formation of cryogenic liquid (Figure 5).

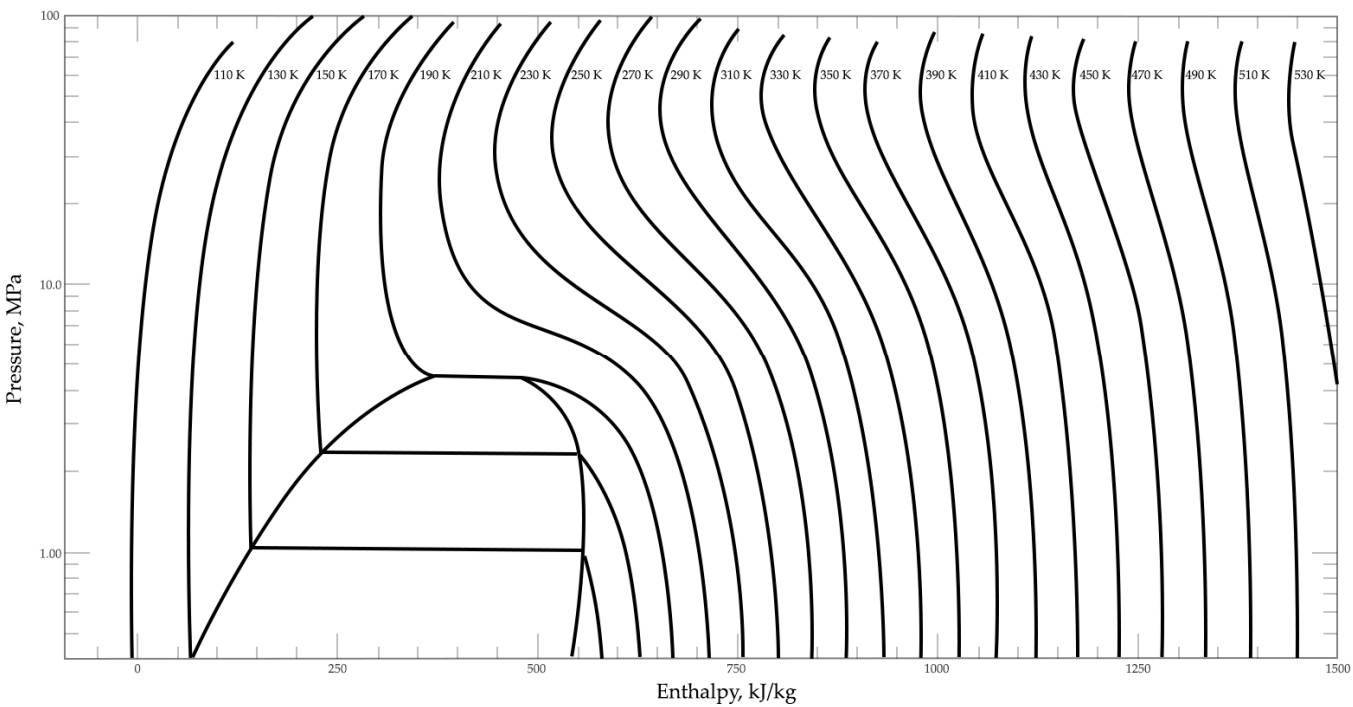

**Figure 5.** Phase diagram of LNG ($CH_4$—99.8%; $C_2H_6$—0.07%; $N_2$—0.13 %).

The area on the left corresponds to a liquid in a single-phase state, and as can be seen from the diagram, the critical point for LNG is located in the area of 190 K at the corresponding pressure. The transportation parameters were chosen in accordance with the area of the diagram located on the left, above the evaporation onset curve. Phase transition temperature is 138.55 K in case of gauge pressure 0.6 MPa.

The three most widely used types of insulation cover were considered: polyurethane foam, aerogel insulation using Cryogel® Z, and vacuum insulation as multi-layer insulation.

Determining the required thickness of the insulating coating is also a difficult task. For its theoretical solution, differential heat balance equations are used. These equations take into account the complex nature of heat transfer during heat flux, convection, and radiation. The application of these equations requires knowledge of the parameters, which can only be determined based on the results of experimental data in each specific case.

Generally, for the three-dimensional formulation of the problem, the heat flux is determined on the basis of Fourier's law:

$$\frac{1}{a} \cdot \frac{\partial t}{\partial \tau} = \frac{\partial^2 t}{\partial x^2} + \frac{\partial^2 t}{\partial y^2} + \frac{\partial^2 t}{\partial z^2}, \tag{1}$$

where $a = \lambda/(\rho \cdot C_p)$; $\lambda$—thermal conductivity coefficient; $\rho$—material density, $C_p$—heat capacity [47].

Within the framework of this study, the data from the standard [37] were used to determine the thickness of the heat-insulating coating.

The linear density of the heat flux through the cylindrical surfaces is calculated by the formula:

$$q_L = \frac{T_{lng} - T_{at}}{R_{int} + R_w + \sum\limits_{i=1}^{n} R_{ins_i} + R_{ext}}, \tag{2}$$

$T_{lng}$—temperature of LNG, $T_{at}$—ambient temperature, $R_{int}$—resistance to heat transfer on the inner surface of the wall of the insulated object, m$^2$·K/W; $R_w$—thermal resistance of the wall of the insulated object, m$^2$·K/W; $\sum\limits_{i=1}^{n} R_{ins_i}$—total linear thermal resistance of n-layer cylindrical insulation; $R_{ins_i}$—linear thermal resistance of the i-layer, m$^2$·K/W; $R_{ext}$—resistance to heat transfer on the outer surface of the thermal insulation, m$^2$·K/W.

Heat transfer resistance and thermal resistance of the walls:

$$R_{int} = \frac{1}{\pi d_{int}^w \alpha_{int}}, \tag{3}$$

$$R_{ext} = \frac{1}{\pi d_{ext}^{ins} \alpha_{ext}}, \tag{4}$$

$$R_{ins} = \frac{1}{2\pi d \lambda_{ins}} ln\frac{d_{ext}^{ins}}{d_{int}^w}, \tag{5}$$

$$R_w = \frac{1}{2\pi d \lambda_w} ln\frac{d_{ext}^w}{d_{int}^w}, \tag{6}$$

$$R_i = \frac{1}{2\pi d \lambda_i} ln\frac{d_{ext}^i}{d_{int}^i}, \tag{7}$$

where $\alpha_{int}$, $\alpha_{ext}$ are the heat transfer coefficients of the inner surface of the wall of the insulated object and the outer surface of the insulation, W/(m$^2$·K); $\lambda_w$, $\lambda_{ins}$, $\lambda_i$—thermal conductivity coefficients of the wall material, of the insulated object of single-layer insulation, insulation of the i-th layer of n-layer insulation, respectively, W/(m$^2$·K); $\delta_w$, $\delta_{ins}$, $\delta_i$—wall thickness of the insulated object, single-layer insulation of the i-layer of n-layer insulation, respectively; $d_{int}^w$, $d_{ext}^w$—inner and outer diameters of the wall of the insulated object; $d_{ext}^{ins}$—outer diameter of the insulation; $d_{ext}^i$, $d_{int}^i$—outer and inner diameters of the i-layer of n-layer insulation.

As mentioned above, heat transfer in vacuum insulation systems is carried out mainly due to radiation.

$$Q = \eta E_0 \sigma_s \cdot 10^8 H\left[\left(\frac{T_2}{100}\right)^4 - \left(\frac{T_1}{100}\right)^4\right], \tag{8}$$

$\eta$—screening efficiency; $E_0$—reduced emissivity of the system of two boundary surfaces; $\sigma_s$—Stefan-Boltzmann constant; $H$—the mutual radiation surface.

Screening efficiency for the cylindrical system:

$$\eta = \left(1 + \frac{E_0}{E_s}r_1 \sum_{m=1}^{n} \frac{1}{r_1 + \frac{m}{n+1}(r_2 - r_1)}\right)^{-1}, \tag{9}$$

$E_s$—reduced emissivity of the system of two adjacent screens, $n$ is the number of screens between shells with outer radii $r_1$ and $r_2$.

In practice, the value of the heat flux is often determined by the formulas for the transfer of heat by thermal conductivity through a cylindrical layer, replacing the coefficient of thermal conductivity in the formula by the value of heat conduction by radiation, $\lambda_{rad}$ [42,47]:

$$q = \frac{2\pi\lambda_{rad}(T_2 - T_1)}{ln\frac{d_2}{d_1}}, \tag{10}$$

At the same time, international standards describe methods for determining the thickness of insulating coatings using simplified formulas based on the following assumptions.

The resistance to heat transfer from the internal environment to the inner surface of the wall for liquid and gaseous media is negligible in comparison with the thermal resistance of the heat-insulating layer and may not be taken into account in practical calculations.

The thermal conductivity of the walls of pipelines made of metal is ten times higher than that of the insulation, so the thermal resistance of the walls can also be neglected without a noticeable decrease in the accuracy of the calculation.

Taking into account the above assumptions, it is possible to use simplified formulas for determining the thickness of the insulating coating for a given decrease (increase) in the temperature of the substance transported by pipelines:

The required total thermal insulation resistance $R = R_{ins} + R_L$ of a pipeline of length L is required to ensure a given temperature increase of the substance transported through it from the initial $t_{lng\_in}$ to the final $t_{lng\_out}$ at the substance flow rate $G$, kg/h, heat capacity $C$, kJ/(kg·°C), $t_{at}$—design ambient temperature is determined from the following expressions:

$$R_{ins} = \frac{3.6K \times L\left(\frac{t_{lng\_in} + t_{lng\_out}}{2} - t_{at}\right)}{G \times C\left(t_{lng\_in} - t_{lng\_out}\right)},\tag{11}$$

$R_{ins}$—thermal resistance of a flat insulation layer, (m²·°C)/W; $K$ is the coefficient of additional losses, which takes into account heat losses through heat-conducting inclusions in heat-insulating structures, occurring due to the presence of fixing devices and supports in them [37].

According to the tables [37], the value of the linear thermal resistance of the outer insulation is $R_l = 0.2$ (m·°C)/W.

Insulating coating thickness:

$$\delta_{ins} = \frac{d_{ext}(B - 1)}{2},\tag{12}$$

where parameter $B = e^{2\pi\lambda_{ins}(R_{ins} - R_{ext})}$; $\lambda_{ins}$ is the coefficient of thermal conductivity of the insulating coating, W/(m²·°C).

The calculation was carried out for the case when the limit of LNG temperature change was set from 113 K to 123 K at an ambient temperature of 293 K. The permissible limit of LNG temperature change is selected taking into account the prevention of liquid boiling processes at a given transportation pressure (in this case, 0.6 MPa) and two types of insulating coating: PU foam and Cryogel® Z.

The vacuum insulated pipeline (VIP) system is introduced as a structure consisting of aluminum foil and glass fiber 0.2 mm thick, with a stacking density of 15 screens per 1 cm for a cylindrical surface, and a total layer thickness of 10 mm [42].

**Table 2.** Material properties.

| | Steel AISI 321 [48] | PU Foam [37] | Cryogel® Z [49] | VIP [42] |
|---|---|---|---|---|
| Density ($\rho$, kg/m³) | 7900 | 70 | 160 | 120 |
| Heat capacity ($Cp$, J/(kg·K)) | 504 | 1400 | 943 | 1030 |
| Thermal conductivity coefficient ($\lambda$, W/(m·K)) | 17 | 0.03 | 0.014 | 0.00011 |
| Thickness, ($\delta$, m) | 0.005 | 0.01 | 0.01 | 0.01 |

## 4. Results

As part of this study, a three-dimensional model of a pipeline with an insulating coating was built in SpaceClaim in ANSYS. Fluent with the symmetry condition, during simulation, we used the "inflation" function to create additional mesh thickening. As a result, the calculations in the near-wall region were more precise. (Figure 6). Solution areas

were discretized as a finite number of elements to obtain solutions numerically using finite volume methods.

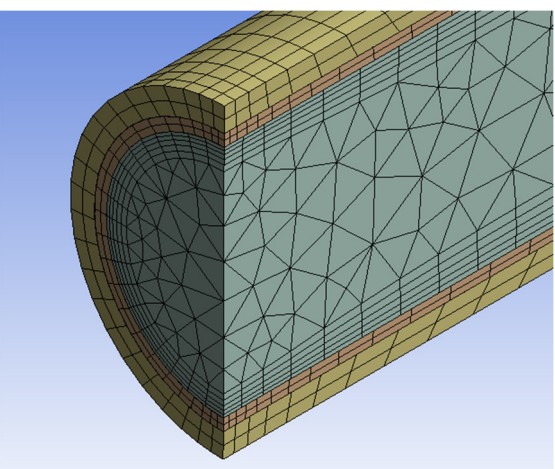

**Figure 6.** Meshing Operation.

As solver settings, the realizable k-epsilon turbulence model with enhanced wall treatment to be used for the LNG pipe flow will enable the energy equation for the simulation. The governing heat transfer equation in ANSYS Fluent takes the following form:

$$\frac{\partial(\rho E)}{\partial t} + \nabla \cdot \left( \vec{v}(\rho E + p) \right) = \nabla \cdot \left( \lambda \nabla T - \sum_j h_j \vec{J}_j + \overline{\overline{\tau}} \cdot \vec{v} \right) + S_h, \tag{13}$$

where $\rho$—density, $E$—total energy, $\nabla$—nabla operator; $p$—pressure; $\lambda$—effective conductivity, $\vec{v}$—overall velocity vector; $\vec{J}_j$—the diffusion flux of species $j$; $h_j$—species enthalpy; $\overline{\overline{\tau}}$—stress tensor, and $S_h$—is the heat of chemical reaction.

Effective conductivity includes the thermal conductivity of the material and the turbulent thermal conductivity, which is defined based on the used turbulence model.

The first three terms on the right-hand side of Equation (13) represent energy transfer due to conduction, species diffusion, and viscous dissipation, respectively [50].

In this case, the transport equation for the realized k-epsilon turbulence model is based on two separate models of the turbulence equation for the turbulence kinetic energy and the energy dissipation rate, which allow both the turbulent length and the time scale to be determined:

$$\frac{\partial(\rho k)}{\partial t} + \frac{\partial(\rho k u_j)}{\partial x_j} = \frac{\partial}{\partial x_j}\left[ \left( \mu + \frac{\mu_t}{\sigma_k} \right) \frac{\partial k}{\partial x_j} \right] + G_k + G_b - \rho\varepsilon - Y_M + S_k, \tag{14}$$

$$\frac{\partial(\rho\varepsilon)}{\partial t} + \frac{\partial(\rho\varepsilon u_j)}{\partial x_j} = \frac{\partial}{\partial x_j}\left[ \left( \mu + \frac{\mu_t}{\sigma_\varepsilon} \right) \frac{\partial\varepsilon}{\partial x_j} \right] + \rho C_1 S\varepsilon - \rho C_2 \frac{\varepsilon^2}{k + \sqrt{v\varepsilon}} + C_{1\varepsilon}\frac{\varepsilon}{k}C_{3\varepsilon}G_b + S_\varepsilon, \tag{15}$$

$$C_1 = max\left[ 0.43; \frac{Sk/\varepsilon}{(Sk/\varepsilon) + 5} \right]. \tag{16}$$

where $k$—turbulence kinetic energy; $\varepsilon$—energy dissipation rate; $u_j$, $v$—components of the flow velocity perpendicular or parallel to the gravitational vector, respectively, $\mu$—dynamic viscosity, $\mu_t$—eddy viscosity; $G_k$—generation of turbulence kinetic energy due to the mean velocity gradients; $G_b$—generation of turbulence kinetic energy due to buoyancy, $Y_m$—the contribution of the fluctuating dilatation in compressible turbulence to the overall dissipation rate, $S_k$, $S_\varepsilon$—user-defined source terms, $\sigma_k$, $\sigma_\varepsilon$, $C_{1\varepsilon}$, $C_{3\varepsilon}$, $C_2$—the model constants, $S = \sqrt{2S_{ij}S_{ij}}$, $S_{ij}$—mean strain rate tensor.

A realizable k-epsilon turbulence model was chosen for further simulation because it takes into account the effect of mean flow distortion on turbulent dissipation and is considered the most proven and widely used [50].

The simulation of the flow of liquefied natural gas in a pipeline section 10 m long was carried out in an insulating coating of PU foam, VIP, and aerogel. The boundary conditions were defined as velocity inlet 1.821 m/s, LNG temperature 113 K, and outlet pressure 0.579471 MPa (calculated, based on hydraulic formulas), while the wall temperature of the insulation was assumed to be equal to the ambient temperature of 293 K at mesh interfaces between LNG and pipe, while pipe and insulation were assumed to be coupled walls.

Figure 7 shows the temperature distribution along the length of the pipeline when PU foam, Cryogel® Z, and VIP are used.

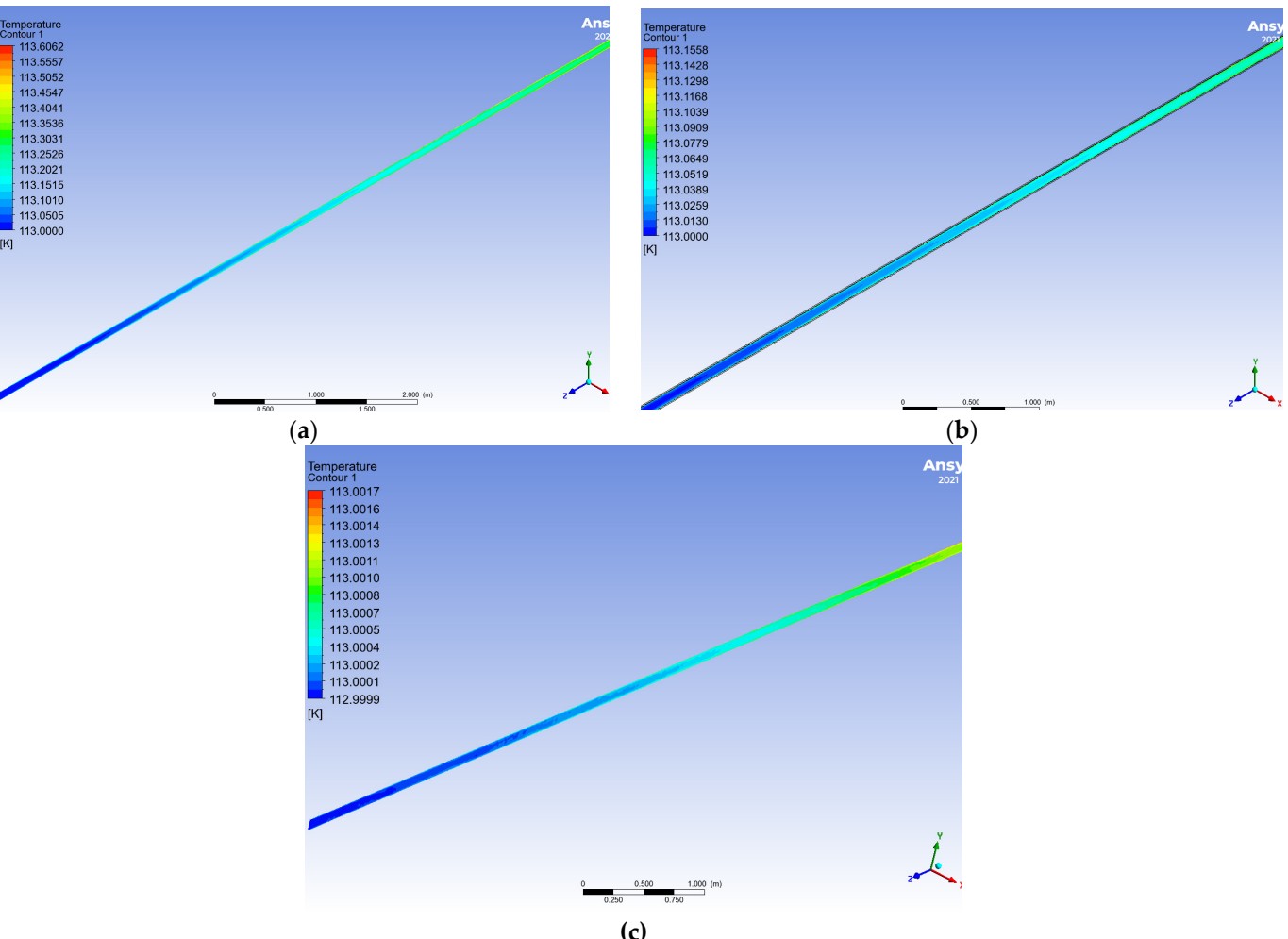

**Figure 7.** LNG flow in the pipeline section with various types of insulation coating. (**a**) polyurethane foam insulation; (**b**) Cryogel® Z; (**c**) VIP.

The LNG flow velocity profile in the cross section of the pipe corresponds to the turbulent flow pattern, which is confirmed in the case of a standard hydraulic calculation for the pipeline (Figure 8).

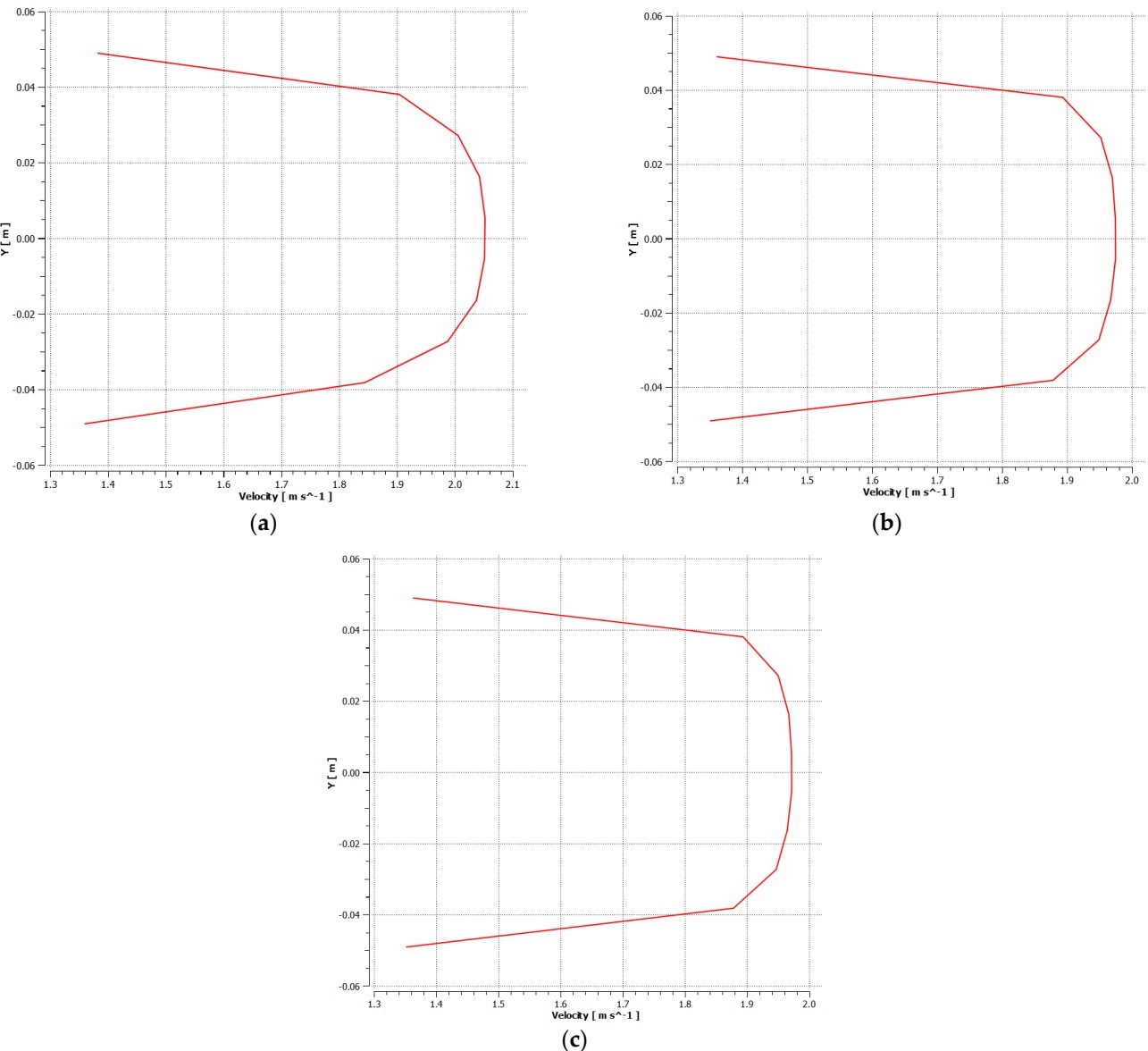

**Figure 8.** LNG flow velocity profile in the cross section of a pipe with various types of insulating coating. (**a**) PU foam insulation; (**b**) Cryogel® Z insulation; (**c**) VIP.

## 5. Discussion

Figure 9 presents a comparative analysis of the LNG temperature distribution. According to the simulation results, it can be said that the temperature of LNG with VIP increases more slowly than for a similar one using aerogels and polyurethane foam insulation.

At the time of the preparation of the article, there was no reliable information on modeling the flow of LNG through pipelines with various types of insulation present. Therefore, to assess the adequacy of the model, the data for similar modeling, which had been performed for liquid nitrogen, were used. The graphs of temperature distribution along the pipeline also correspond to the theoretical expectations presented in other studies [40,41].

Due to the complexities of using VIP, as mentioned in Section 2.2, the LNG flow in pipelines with different thicknesses of polyurethane foam insulation (0.01 m; 0.05 m; 0.1 m) was additionally simulated in order to reduce heat gains to the level of insulation provided for VIP (Figure 10).

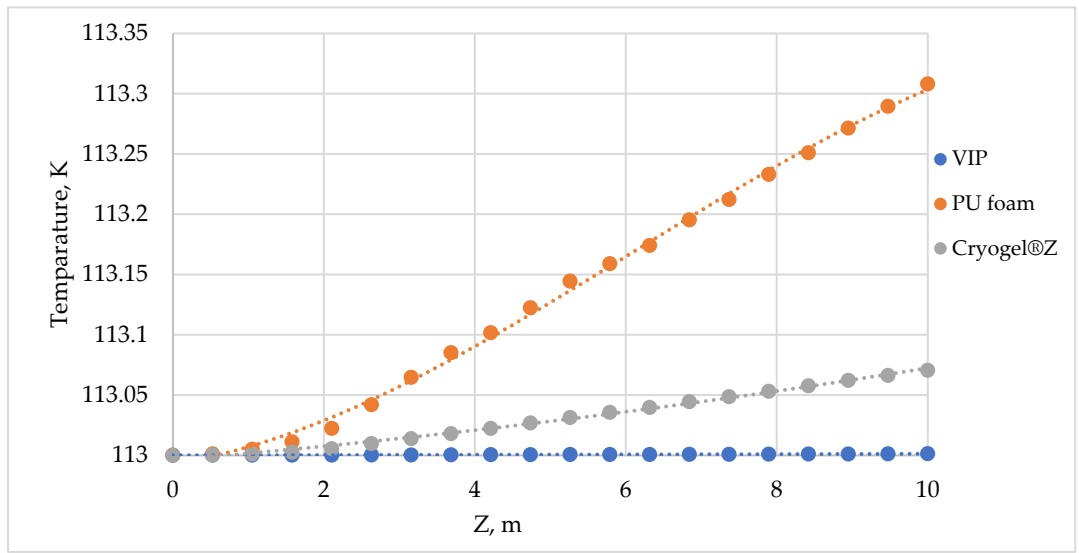

**Figure 9.** Asymmetrical temperature distribution along the pipeline.

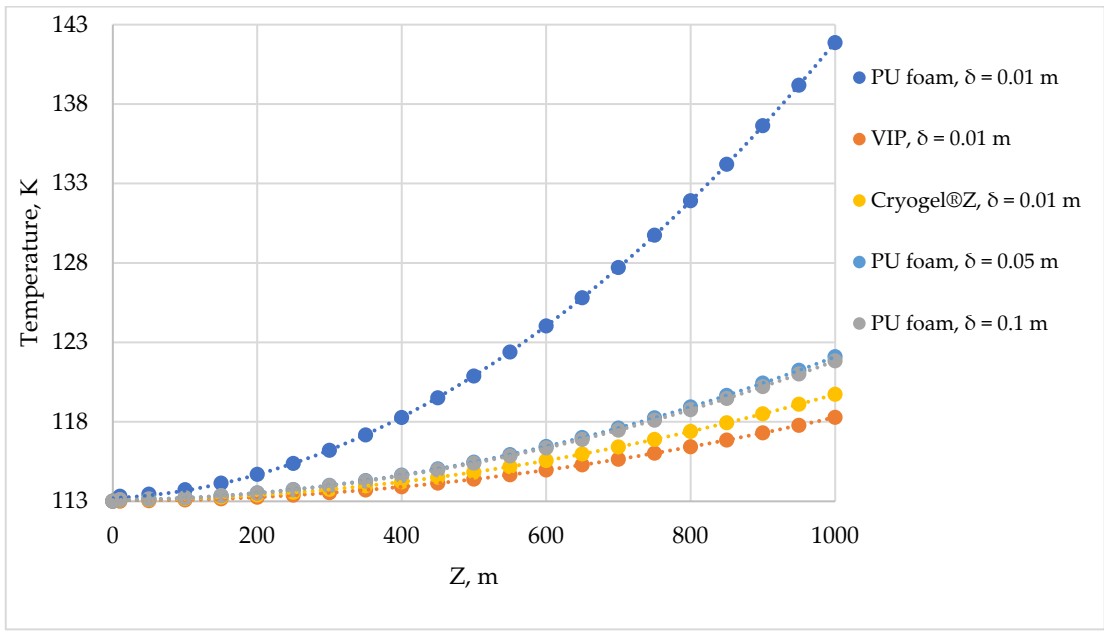

**Figure 10.** Asymmetrical temperature distribution along the pipeline with different thickness of PU foam insulation.

Due to the fact that the use of CFD simulation for long-length pipelines (more than 100 m) is limited, the simulation results were approximated by curve resolution and then extrapolated to the length of 1000 m. Extrapolation was carried out by an equation of the form:

$$T(x) = ae^{-bx} + c, \tag{17}$$

where $a = T_{in} - c$; $b$, $c$—coefficients of best fit.

As a result of the simulation results extrapolation, the LNG temperature in a pipe with a thickness of PU foam insulation of 0.01 m exceeds the LNG phase transition temperature (138.55 K), which leads to two-phase flow formation and destabilization of flow.

An additional increase in the thickness of the PU foam layer up to 0.05 m for a pipe with an external diameter of 0.108 m provides a significant increase in the insulation level, while a further increase in the thickness up to 0.1 m no longer has a significant effect.

Further, it is possible to provide an insulation level comparable to VIP; the difference between the two types of insulation is 3.81 K per 1000 m of pipe. Regarding the light weight of PU foam, the loading arising from the weight of such an insulating coating does not have a significant effect on the stress-strain state in comparison with the loading from the weight of the LNG and the pipeline material itself, which is observed from the low specific gravity of the material.

As an alternative to the proposed solutions, the authors consider the possibility of using polymer materials. To date, the production of polymer materials is intensively rising, new materials are appearing that, in their strength characteristics, are comparable to widely used steels, and there is a growing interest in research into the behavior of polymers at cryogenic temperatures [51–53].

Ultra-high molecular weight polyethylene is a type of polyethylene (UHMWPE) with a molecular weight of $1 \times 10^6$ and above. Advanced researchers consider it a promising structural material for construction due to its great material performance, including wear resistance, strength, and high impact strength at cryogenic temperatures (up to $-196 \,^\circ$C) [54].

The simulation of LNG flow in a UHMWPE pipeline with an external diameter of $d_{ext}$ = 0.11 m and a wall thickness $\delta$ = 0.01 m (density: 930 kg/m$^3$, heat capacity: 1900 J/(kg·K), thermal conductivity coefficient: 0.3 W/(m·K)) with a PU-foam insulation coating was performed in a similar manner. Figure 11 depicts the LNG temperature distribution along the length of the UHMWPE pipeline when different thicknesses of PU foam were used.

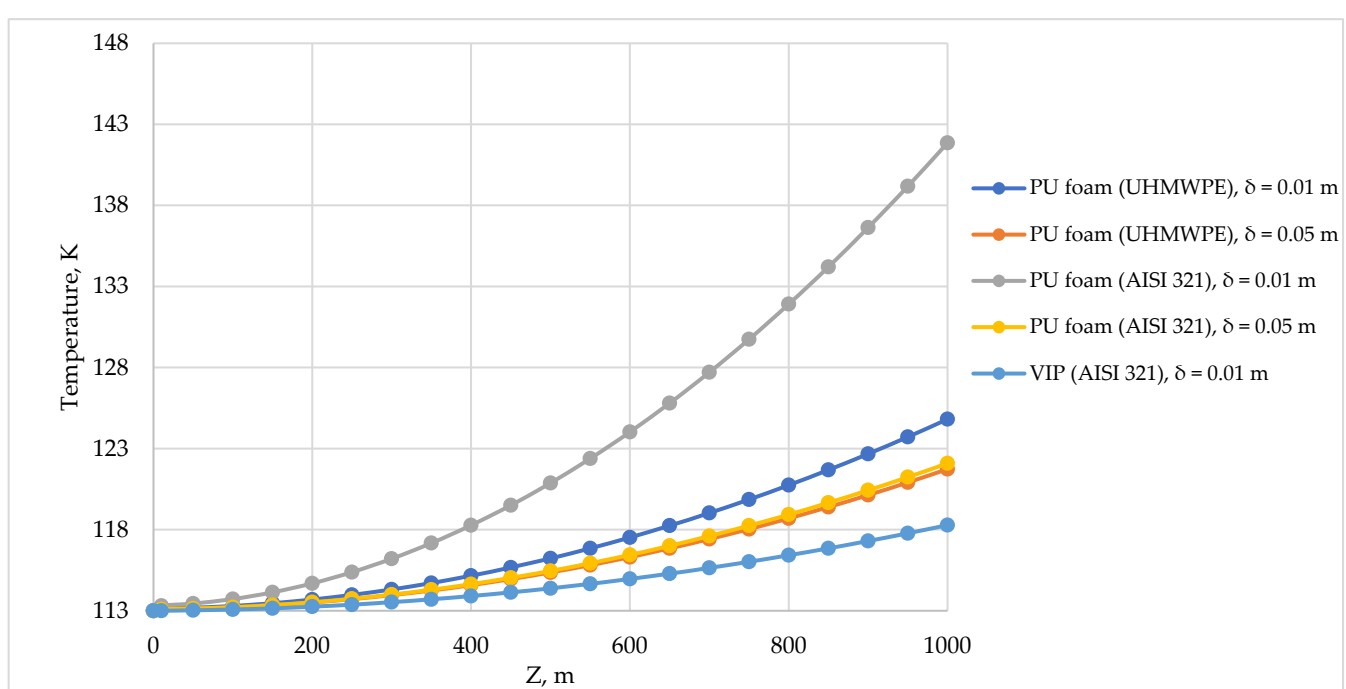

**Figure 11.** Asymmetrical temperature distribution along the UHMWPE pipe with different thickness of PU foam insulation.

With an increase in the thickness of UHMWPE pipe's insulation to 0,05 m, the level of temperature distribution in the pipeline from UHMWPE and steel is almost the same, which is an interesting result, since there is a significant difference in the thermal conductivity coefficient of UHMWPE $\lambda$ = 0.3 W/(m·K) and steel $\lambda$ = 17 W/(m·K).

Table 3 compares the maximum temperature in the pipe with different dimensional ratios between the thickness of the pipe and the insulating coating.

**Table 3.** Maximum temperature of LNG in the pipeline at different thicknesses of the PU-foam insulation.

|  | **Steel AISI 321** | **UHMWPE** |
|---|---|---|
| Insulation thickness ($\delta_{ins}$, m) | 0.01 | |
| Maximum temperature (*Tmax*, K) | 141.861123 | 124.8126961 |
| Insulation thickness to pipeline thickness ratio | 2 | 1 |
| Insulation thickness ($\delta_{ins}$, m) | 0.05 | |
| Maximum temperature (*Tmax*, K) | 122.0924843 | 121.7345773 |
| Insulation thickness to pipeline thickness ratio | 10 | 5 |

Apparently, an increase in the thickness of the insulating coating relative to the thickness of the pipe wall leads to the fact that the influence of the thermal conductivity of the pipe material decreases; therefore, the geometry factor (the ratio between the pipe wall thickness and the insulation thickness) and the thermal conductivity of the insulating material become the determining parameters.

## 6. Conclusions

The article considers the main technical solutions for the construction of LNG process pipelines and their insulation systems.

The ANSYS Fluent software package simulated a section of a pipeline pumping liquefied natural gas with three types of insulation coating: PU foam, Cryogel® Z, and VIP.

Based on the results of the obtained model, a comparative analysis of temperature change in a pipeline made of AISI 321 cryogenic steel with various types of insulating coating was performed. It was found that the rate of LNG temperature increase in a pipeline with polyurethane foam insulation is greater than that of Cryogel® Z and VIP. At the same time, by increasing the thickness of the PU foam insulating coating, a similar effect of reducing heat gains can be achieved.

The UHMWPE is considered an alternative material for pipeline construction is considered. UHMWPE has a great combination of strength properties that might be used in the cryogenic industry.

The temperature of LNG increases more slowly in a UHMWPE pipe than in a similar one made from steel with an insulating coating thickness of up to 0.05 m; further, with an increase in the thickness of the insulating coating, the influence of the scale factor becomes noticeable.

Despite the rapid development of the liquefied natural gas industry, there is not enough research on the performance of liquefied gas when being moved through pipelines under pressure, which is probably due to the complexity of both theoretical and experimental studies.

In spite of the above-mentioned factors hindering the development of the LNG industry, its increasing role in the global energy mix seems to be the most promising scenario for the development of the energy sector in the near future.

**Author Contributions:** Conceptualization, supervision, and writing—editing I.S.; conceptualization, methodology, software, and writing—original draft preparation E.K. All authors have read and agreed to the published version of the manuscript.

**Funding:** This research received no external funding.

**Institutional Review Board Statement:** Not applicable.

**Informed Consent Statement:** Not applicable.

**Data Availability Statement:** The data presented in this study are available on request from the corresponding author.

**Conflicts of Interest:** The authors declare no conflict of interest.

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
