# Peer review of "The LNG Flow Simulation in Stationary Conditions through a Pipeline with Various Types of Insulating Coating"

_fluids, doi:10.3390/fluids8020068_

Round 1

Reviewer 1 Report

The authors offer an excellent overview of various insulation strategies for cryogenic pipelines in this work which includes key examples of industrial applications. This forms the basis of a useful document worthy of publication. While my comments are mostly minor, at present I believe there is a gap between the industrial scales (on the order of km) that are discussed in review, and the simulations (1 m) that are conducted. Before publication, the authors should rationalise this gap through a combination of additional discussion and/or modelling work as discussed below.

Major points:

The extensive literature review conducted by the authors gives details on several pipelines that are on the order of 100s of meters to km in length. Their simulations in sections 4 and 5 are conducted on a 1 m line length and show temperature fluctuations that would not likely be measurable in the field. More work is needed to bridge the gap between the industrial scales discussed and the scale of the simulation. This may take the form of additional simulations and/or additional discussion to rationalise the overall body of work.

The authors give a detailed description of the model that they employ in CFD. How does this compare to a simple 1D calculation assuming homogeneous fluid temperature based on m*c_p*dT = U*A*dT? Are significant differences apparent between the 1D (bulk heating) case and the 3D (wall layer heating) cases? Some discussion of this is merited – particularly if it is found that use of a 1D model will underestimate the likelihood of boiloff by a significant margin (based on the % difference in temperature over a given fluid residence time).

Minor points:

P2 Li 49: While the authors’ meaning is clear, “water” hammer is perhaps not the ideal term here. Perhaps “fluid” or “liquid” hammer instead?

P2 Li 68-76: As the generally stated purpose of LH2 use is a reduction in greenhouse gas emissions, problems with i) containment; ii) boil off during transport and iii) its 100 year global warming potential on the order of 10 may also merit mention.

Section 3: Could the authors be consistent in their use of symbols for temperature: equations 1 and 2 use a lower case t, whereas 8 uses a capital T (the latter is preferred).

Overall: a nomenclature section summarising all symbols and units used throughout would be useful.

P10, li 363-364: “The calculation was carried out for the case when the limit of the LNG temperature change was set from 103 to 113 K …” I found this sentence confusing. The inlet boundary condition for the simulation was set to 113 K, so we would expect only temperatures greater than 113 K in these simulations – what’s the relevance of introducing the lower bound?

P10 li 365-366: It may also be useful to talk about the limits/boundaries of the simulation by including a phase diagram in the region of interest so that the reader can appreciate the operating range (pressure, temperature) that the pipe must be confined to.

P7 li 291: Which equation of state is REFPROP using here? GERG 2008 or something else?

What do the dotted lines represent in figures 8,9,10? Are these equations, lines of best fit or simply a guide?

Reviewer 2 Report

The manuscript presents an overview of the main technical solution for the development and construction of the LNG pipeline. A cryogenic pipeline with different insulation material and thickness is also simulated using ANSYS Fluent. The manuscript was well written and clearly to understand. 

Section 2 discusses the world experience in the construction of liquefied gas pipelines. It gives reader a comprehensive design and development for cryogenic pipeline. 

Regarding the section 3, the cryogenic pipeline is simulated with different insulation material and thickness. Figure 9 shows that increasing insulation layer from 1cm to 20cm only make outlet temperature decrease 0.01K. I believe that 0.01K is meaningless comparing up to 20cm increase in the insulation thickness. The same comment is also for Figure 10. To clear the impact of thickness or material of insulation layer, a much longer pipeline simulation may necessary. 

Line 458-459, page 15: Please use the same length unit in the whole article. Because almost dimension in this study is relatively small, I recommend author to use cm instead of m. 

Finally, the reviewer thinks that section 3 should be improved to insight into the importance of the insulation material and thickness. For example, increase the length of simulation pipeline to see more effect of insulation thickness. The current result presented in section 3 is not much valuable from the reviewer's opinion.

Round 2

Reviewer 1 Report

The authors have addressed my previous comments on the manuscript and I believe it may be published in its current form.

Reviewer 2 Report

The paper was well-revised. It is recommended to publish the article in the MDPI Fluids Journal. 

Reviewer has an additional suggestion on the Fluent model. In the future with pipeline calculation, I recommended to used 2D model with X-axisymmetric. Because the geometry symmetry, the 2D model would give reasonable results while reducing the number of meshes and thereby the calculation time a lot.